# Identification of the Prognostic Biomarkers CBX6 and CBX7 in Bladder Cancer

**DOI:** 10.3390/diagnostics13081393

**Published:** 2023-04-11

**Authors:** Xinxin Li, Lili Li, Xi Xiong, Qihui Kuang, Min Peng, Kai Zhu, Pengcheng Luo

**Affiliations:** 1Department of Urology, Wuhan Third Hospital and Tongren Hospital of Wuhan University, Wuhan 430060, China; 2Central Laboratory, Renmin Hospital of Wuhan University, Wuhan 430060, China; 3Department of Urology, Wuhan Third Hospital, School of Medicine, Wuhan University of Science and Technology, Wuhan 430060, China; 4Department of Oncology, Renmin Hospital of Wuhan University, Wuhan 430060, China; mpeng320@whu.edu.cn; 5Department of Gastroenterology, Renmin Hospital of Wuhan University, Wuhan 430060, China

**Keywords:** chromobox proteins, bladder cancer, biomarkers, immune environment, urothelial cancer

## Abstract

Background: Chromobox (CBX) proteins are essential components of polycomb group proteins and perform essential functions in bladder cancer (BLCA). However, research on CBX proteins is still limited, and the function of CBXs in BLCA has not been well illustrated. Methods and Results: We analyzed the expression of CBX family members in BLCA patients from The Cancer Genome Atlas database. By Cox regression analysis and survival analysis, CBX6 and CBX7 were identified as potential prognostic factors. Subsequently, we identified genes associated with CBX6/7 and performed enrichment analysis, and they were enriched in urothelial carcinoma and transitional carcinoma. Mutation rates of TP53 and TTN correlate with expression of CBX6/7. In addition, differential analysis indicated that the roles played by CBX6 and CBX7 may be related to immune checkpoints. The CIBERSORT algorithm was used to screen out immune cells that play a role in the prognosis of bladder cancer patients. Multiplex immunohistochemistry staining confirmed a negative correlation between CBX6 and M1 macrophages, as well as a consistent alteration in CBX6 and regulatory T cells (Tregs), a positive correlation between CBX7 and resting mast cells, and a negative correlation between CBX7 and M0 macrophages. Conclusions: CBX6 and CBX7 expression levels may assist in predicting the prognosis of BLCA patients. CBX6 may contribute to a poor prognosis in patients by inhibiting M1 polarization and promoting Treg recruitment in the tumor microenvironment, while CBX7 may contribute to a better prognosis in patients by increasing resting mast cell numbers and decreasing macrophage M0 content.

## 1. Introduction

Bladder cancer (BLCA) is the most common form of urothelial cancer worldwide. It ranks first in the incidence of genitourinary cancer in China, while in the West, its incidence ranks second to prostate cancer [1,2]. The incidence rate of BLCA increases with age and reaches a peak at 50–70 years old [3,4]. The etiology of BLCA includes a variety of genetic and environmental factors [5]. Although accumulated studies have explored the mechanisms of development and progression for BLCA, the mechanism of BLCA in the tumor microenvironment remains under investigation, and the treatment options for BLCA are also limited [6]. Understanding the underlying pathogenesis of BLCA in the tumor microenvironment could shed light on identifying diagnostic biomarkers and pave the way for advanced treatment methods [7].

Cancer cells and immune cells constitute the tumor microenvironment (TME) of BLCA, which has been confirmed as a crucial regulator of the development of BLCA. Single-cell RNA sequencing of the TME showed the single-cell-level immune cell landscape [8] and uncovered unique clusters of immune cells linked with tumors and signature genes for diverse types of immune cells [9]. Multiple studies have shown that entinostat, a selective class I HDAC inhibitor, induces an anticancer response through immunological editing of tumor neoantigens and modification of the tumor immune microenvironment [10]. Programmed cell death 1 (PD-1) and PD-1 ligand (PD-L1) checkpoint blockade inhibited tumor growth in a CD4+ T-cell- and macrophage-dependent manner in a urinary bladder matrix-created immune microenvironment [11].

Polycomb group (PcG) proteins are essential gene regulators because they facilitate the persistent transmission of cell state inheritance [12]. PcG protein-mediated aberrant epigenetic regulation has been widely studied in relation to a variety of cancer types. CBX family proteins, which are essential PcG constituents, govern progression and carcinogenesis for numerous types of cancer by preserving tumor suppressors and undifferentiated cancer stem cell states [13]. Human genomes include eight members of the CBX protein family, including CBX1-8. All of them contain conserved N-terminal chromodomains. Based on variations in molecular structure, CBXs may be categorized into two groups: the heterochromatin protein 1 subfamilies CBX1/3/5 and the polycomb subfamilies CBX2/4/6/7/8 [14]. CBXs have important functions, including heterochromatin regulation, polycomb repressive complex 1 (PRC1) recruitment and stability to different regions of chromatin, and the regulation of PRC1 binding to nucleosomes. The HP1 group is related to transcriptional silencing by binding the H3K9me2/me3 marks [15]. Simultaneously, the major functional member Pc repressive complex 2 (PRC2) of the Pc group modifies H3K27me2/3 to transcriptionally repress [16]. Aberrant expression of the CBX family in BLCA is closely associated with its prognostic value. CBX6 was discovered to be increased in hepatocellular carcinoma, which is related to decreased patient survival, according to previous research [16]. Low expression of CBX7 is accompanied by shorter overall survival [17]. In the progression of BLCA, the connection between CBX proteins and tumor-infiltrating immune cells remains unknown.

In this research, the connection between CBX expression and BLCA patient prognosis was investigated, and CBX6 and CBX7 were identified as promising prognostic genes. We analyzed the gene mutations of individuals with high and low expression of CBX6 and CBX7, evaluated immune cell infiltration by different immune algorithms in tumor tissue, and then verified the immune infiltration results. Systematic investigations of the function of CBXs in BLCA will provide information for future investigation of the molecular processes behind CBXs in the development of BLCA and provide predictive biomarkers and immunotherapeutic targets for clinical cohort studies.

## 2. Materials and Methods

### 2.1. ONCOMINE Database

The ONCOMINE database (http://www.oncomine.org) accessed on 1 January 2022. integrates cancer microarray data to provide genome-wide expression analyses [18]. In this study, CBX transcriptional expression was extracted. Student’s *t*-test was performed to investigate the difference between normal and malignant transcriptional expression.

### 2.2. The Cancer Genome Atlas Database

The Cancer Genome Atlas (TCGA, https://www.cancer.gov/tcga) accessed on 1 January 2022 is a groundbreaking cancer genome project that has sequencing data and clinical information on various forms of cancer [19]. RNA-seq data and corresponding clinical data of BLCA were downloaded, including 419 BLCA samples and 11 normal samples. The FPKM-formatted RNA-seq data were log2 transformed. The R program ggplot2 was used to visualize the mRNA expression of the samples. The R package pROC was applied to the receiver operating characteristic (ROC) curve.

### 2.3. Kaplan–Meier Plotter Database

CBX mRNA expression in BLCA was evaluated for its predictive value using Kaplan–Meier plotter (http://kmplot.com/analysis) accessed on 3 January 2022 [20]. According to the autoselect best cut-off values of CBX mRNA expression, BLCA patients were divided and further validated by KM survival curves.

### 2.4. TISIDB Analysis

The TISIDB database (http://cis.hku.hk/TISIDB) accessed on 3 January 2022 consists of 988 documented immune-related genes in cancer and studies the relationship between genes and immune cell infiltration by analyzing high-throughput screening information and genomes, transcriptomics, and clinical information [21]. In the present study, we examined the relationships among CBX expression, clinical data, and subtype.

### 2.5. The Human Protein Atlas Database

The Human Protein Atlas (https://www.proteinatlas.org) accessed on 3 January 2022 is a great online resource that includes immunohistochemistry-based expression cell line and tissue information for most identified genes [22]. In the current research, the protein expression levels of CBXs were compared between human normal and BLCA samples by obtaining an immunohistochemical image.

### 2.6. GeneMANIA

GeneMANIA (http://www.genemania.org) accessed on 5 January 2022 is a multifunctional website for providing gene function, protein–protein interaction, links between genes and datasets, functionally comparable genes, and the common characteristics of related genes [23].

### 2.7. Metascape

Metascape (https://metascape.org) accessed on 8 January 2022 is a reliable online resource for gene annotation and enrichment analysis [24]. In this research, coexpressed genes and CBX functions were investigated in Metascape using GO and KEGG.

### 2.8. TIMER

TIMER (https://cistrome.shinyapps.io/timer) accessed on 9 January 2022 is a database for systematically studying the immunological infiltration of diverse immune cells and their clinical effects across a variety of cancer types [25]. In the “Survival” module, the clinical importance of CBX expression levels was evaluated using a multivariable Cox proportional hazard model.

### 2.9. Mutation Data, Heatmaps, and Analysis of Immune Checkpoints and m6A in CBXs

Mutation data from the TCGA database and display of mutations were evaluated with the package maftools. The R programming language and the pheatmap software package were used to generate heatmaps of DEGs, infiltrating immune cells, and variations in immune function. The reshape2 package was used to analyze the differentially related genes of m6A and immune checkpoints between the CBX6 and CBX7 high- and low-expression groups.

### 2.10. Tumor-Infiltrating Immune Cell Profile

We calculated immune cell infiltration in all tumor samples using seven methods (EPIC, XCELL, MCPCOUNTER, QUANTISEQ, CIBERSORT-ABS, CIBERSORT, and TIMER); the research only included samples with *p* < 0.05.

### 2.11. Cell Lines

BeNa Culture Collection (Beijing, China) supplied the normal urothelial cell line (SV-HUC-1) and the bladder cancer cell lines (5637 and T24). All cells were cultured in DMEM F-12 medium with 10% FBS (Gibco) at 37 °C and 5% carbon dioxide. All cell lines were mycoplasma-free by mycoplasma testing (Servicebio, #G1900-50T, Wuhan, China).

### 2.12. Western Blotting

To extract the complete cell proteins, RIPA lysis buffer was applied. Using a BCA protein quantification kit (Servicebio #G2026-200T), the concentrations of total proteins were measured. We loaded 15 ug of soluble protein on each gel well (SDS-PAGE gel preparation kit, Servicebio). PVDF membranes were transferred with proteins. PVDF membranes were treated overnight at 4 °C with particular primary antibodies after being blocked with 5% milk in TBST. The membranes were subsequently treated with a secondary antibody for two hours at room temperature. The membranes were then analyzed using gel imaging equipment and chemiluminescence (Bio-Rad Laboratories, Hercules, CA, USA). Anti-rabbit CBX6 and CBX7 antibodies were obtained from Affinity (catalogue numbers #AF0411 and #DF13257, respectively, INERLAB Co., Ltd., Taipei, Taiwan). The Goat Anti-Rabbit IgG (H + L) Fluor594-conjugated (S0006, Affinity) was used as secondary antibody.

### 2.13. Statistical Analysis

GraphPad Prism 8 (San Diego, CA, USA) was utilized for statistical analysis, and various group means were compared using ANOVA (one-way variance analysis). When the *p*-value was less than 0.05, statistical significance was established. The data are provided as the mean standard measurement error.

### 2.14. Multiple Immunohistochemistry

mIHC staining was performed on BLCA tissue samples. For both brightfield and fluorescence multiplex immunohistochemistry, 4 µm tissue slices were employed. Before inhibiting endogenous peroxidase and incubating the antibody, in an autoclave, tissue sections were dewaxed and treated with Tris-EDTA pH 7.8 antigen retrieval solution for 5 min at 121 °C. To perform multiplex fluorescence immunohistochemistry (mIHC), antibody staining had to be carried out in a series of sequential steps. These procedures comprised inhibiting peroxidase, applying the primary antibody, detecting with a secondary horseradish peroxidase (HRP)-conjugated antibody, detecting with a fluorescent dye, and using a microwave to remove attached antibodies. In each experiment, three different antibodies were stained using three different circulations. The slides were mounted in an antifade solution after being counterstained with diamidino-2-phenylindole (DAPI). Bladder cancer tissue sections were obtained from Shanghai Outdo Biotech Co., Ltd. (Outdo Biotech, #HBlaU050CS01). Antibodies used in multiple immunohistochemistry were CBX6 (Affinity, #AF0411), CBX7 (Affinity, #DF13257), KIT (Proteintech, #18696-1-AP, Rosemont, IL, USA), CD14 (Servicebio, #GB11254), CD86 (Proteintech, #13395-1-AP), and CD25 (ABclonal, #A23153, Woburn, MA, USA).

## 3. Results

### 3.1. Expression Levels of CBXs

CBX mRNA expression was discovered in numerous cancer types (Appendix A). Eight members of the CBX clan, CBX1/2/3/4, were significantly expressed at a high level, whereas CBX6/7 was significantly downregulated in most cancer types, such as BLCA. These findings were consistent with observations of diverse cancer types, indicating that the CBX family has a conserved role across tumor types (Appendix A).

To confirm the mRNA expression profiles of CBXs in BLCA, we downloaded RNA-seq data from TCGA, comprising 19 normal tissues and 411 BLCA tissues, and created a heatmap of CBX family gene expression (Figure 1A). mRNA expression of CBX1/2/3/4/8 in BLCA samples was considerably increased; however, mRNA expression of CBX6/7 in BLCA samples was considerably decreased in unpaired and paired analyses (Figure 1B,C). We further compared the expression of CBX mRNA between 408 BLCA and 19 normal tissues using the online database UALCAN. The mRNA expression levels of CBX1/2/3/4/8 were higher in BLCA tissues. However, CBX6/7 mRNA expression was significantly lowered in tumor samples (Appendix A), which was consistent with prior findings [17].

We further investigated the expression patterns of the CBX proteins in BLCA (Appendix A). The protein expression levels of CBX1/2/3/4/5/8 were dramatically increased in BLCA tissues. Lower expression levels of CBX6/7 proteins were observed in BLCA tissues. The protein expression levels of CBX1/2/3/4/6/7/8 corresponded to the mRNA expression variations.

### 3.2. Association of CBX mRNA Expression with Immune Subtypes in BLCA Patients

CBX mRNA expression was significantly associated with five immunological subtypes evaluated in TISIDB (Figure 2). CBX1/2/3/8 mRNA expression levels were considerably higher in the wound healing subtype (C1) than in the other subtypes (Figure 2A–C,H), but the levels of CBX6/7 mRNA were considerably greater in the inflammatory subtype (C3) and lower in the IFN-gamma dominating subtype (C2) (Figure 2F,G).

### 3.3. Prognostic Factors Associated with Mortality of BLCA

We further assessed risk factors for the mortality of 406 BLCA patients in TIMER. The risk factors for mortality in Table 1 were evaluated by a multivariable Cox proportional hazard model. The mRNA expression of CBXs and clinical factors were used for analysis in the multivariable Cox regression model (Table 1). Multivariate analysis showed that four variables were estimated as prognostic factors for mortality in BLCA: age, (HR = 1.035, *p* = 0.000), CBX5 (HR = 1.328, *p* = 0.014), CBX6 (HR = 1.334, *p* = 0.001), and CBX7 (HR = 0.696, *p* = 0.000), which were dramatically related to the clinical outcome of BLCA patients (Table 1). The ROC curves for each CBX gene are shown in Figure 3. CBX7 has an AUC of 0.9631, which is the highest among all others. In addition, the AUC for CBX3/8 was more than 0.9, and the AUC for CBX1/2/4/6 was more than 0.75, while the AUC of CBX5 is only 0.5299. These results indicated that the prognostic index of CBX7 shows the most remarkable efficacy in stratifying patients. The results of multivariate Cox regression analysis were verified by the Assistant of Clinical Bioinformatics website (Appendix A).

### 3.4. Prognostic Value of CBXs in BLCA Patients

To assess the prognostic importance of CBX mRNA expression levels in the development of BLCA, we studied the relationship between clinical data and CBX expression levels in BLCA patients. CBX7 mRNA expression was related to a greater overall survival rate (log-rank *p* = 0.0161) in BLCA patients, as determined by microarray analysis (Figure 4G). The mRNA expression level of CBX6 was correlated with a worse overall survival rate (log-rank *p* = 0.0249) (Figure 4F). Therefore, CBX6/7 mRNA expression was dramatically associated with BLCA patient prognosis as biomarkers for BLCA patient survival prediction.

### 3.5. Functional Enrichment Analysis of CBXs

CBX6 and CBX7 mRNA expression in BLCA was evaluated for their prognostic value using the Kaplan–Meier plot (Figure 5A,B). The prediction results were consistent with the results shown in the TISIDB database (Figure 4F,G). Coexpression neighbor gene analysis of CBX6 and CBX7 was performed using GeneMANIA to search for potential interactions (Figure 5C,D). The functions of CBX6 and CBX7 and their neighboring genes were analyzed using Metascape. Our results showed a summary of the enrichment analysis in the DisGeNET section of Metascape. CBX6 and CBX7 play a critical role in bladder cancer formation, as shown by the results of enrichment analysis based on their coexpressed genes, which revealed that they were associated with urothelial carcinoma and carcinoma and transitional cells. Analyses related to CBX6 include diffuse intrinsic pontine glioma, urothelial carcinoma, MIXED LINEAGE LEUKEMIA, carcinoma, transitional cell, lymphoma, follicular, gastrointestinal stromal tumors, and eosinophil count procedures (Figure 5E). CBX7-related assays included central serous chorioretinopathy, urothelial carcinoma, MIXED LINEAGE LEUKEMIA, carcinoma, transitional cell, lymphoma, follicular, and eosinophil count procedures (Figure 5F).

### 3.6. Mutation Analysis

The median level of CBX6 mRNA expression was utilized to divide BLCA patients into two groups with high and low expression, and mutations were identified for both groups. The findings demonstrated that mutations of the top fifteen genes in each group, including TTN and the tumor suppressor gene TP53, were more prevalent in the group with high CBX6 expression (Figure 6A,B). The same method was used to classify BLCA patients using median values of mRNA expression of CBX7. The findings demonstrated that the mutation frequency of TTN and the tumor suppressor gene TP53 was lower in the CBX7 high-expression group, as was the mutation frequency of genes (Figure 6C,D).

### 3.7. Relationship of CBXs and m6A

Histone functions are often accompanied by methylation. We investigated the methylation changes between the high- and low-expression groups of CBX6 and CBX7 and discovered that the methylation of CBX6 and CBX7 was considerably elevated in the bladder cancer group (Figure 7A,B). The m6A-related genes that differed between the CBX7 high- and low-expression groups were METTL14, YTHDC1, FTO, YTHDC2, YTHDF1, YTHDF2, METTL3, RBM15, ALKBH5, and ZC3H13 (Figure 7C). The m6A-related genes that differed between the two CBX7 high- and low-expression groups were METTL14, YTHDC1, FTO, YTHDC2, YTHDF1, HNRNPC, METTL3, and ZC3H13 (Figure 7D).

### 3.8. Immune Cell Infiltration of CBX6 and CBX7 in BLCA Patients

We assessed the association between CBX6, CBX7, and immune cell infiltration using seven methods: EPIC, XCELL, MCPCOUNTER, QUANTISEQ, CIBERSORT-ABS, and CIBERSORT (Figure 8A and Figure 9A). Concurrently, we also evaluated the variations in immune function between the CBX6 and CBX7 high- and low-expression groups. Type I IFN response immune function was active in patients in the CBX6 high-expression group, while inflammation promotin, cytolytic activity, APC coinhibition, type I IFN repair, parallelism, MHC class I, and HLA immune function were active in patients with low CBX6 expression (Figure 8B). Type II IFN response immune function was active in patients with high CBX7 expression, while MHC class I and APC coinhibition immune function was active in patients with low CBX7 expression (Figure 9B). The immune checkpoints that differed between the CBX6 high- and low-expression groups were TNFRSF18, CD200, TNFSF15, CD28, LGALS9, CD44, TNFRSF8, CD160, ADORA2, AIDO1, CD276, TNFRSF25, BTNL2, CD40, ICOSLG, and VTCN1 (Figure 8C). Similarly, the immune checkpoints that differed between the CBX7 high- and low-expression groups were PDCD1LG2, CD27, TNFSF15, CD28, LGALS9, CD44, TNFRSF8, CD160, ADORA2A, CD200R1, IDO2, CD274, BTLA, BTNL2, CD40 TNFRSF14, ICOSLG, CD40LG, and LAG3 (Figure 9C).

The CIBERSORT method was used to quantify the proportion of immune-related subgroups that infiltrated tumors. Consequently, the association between CBX6 and the immunological milieu may be elucidated further, and 21 immune cell profiles were also generated from BLCA data. Five types of immune cells had clearly distinct expression between the CBX6 high- and low-expression groups (Figure 10A). There was a total of six immune cells associated with the expression of CBX6. Following the intersection of differentially expressed immune cells and related cells, four immune cells linked with the tumor microenvironment (TME) were eventually identified (Figure 10B). Tregs and naïve B cells had a positive correlation with CBX6 expression, while M1 macrophages and resting NK cells had a negative correlation (Figure 10C).

In the same way, in the CBX7 high- and low-expression patient groups, we obtained six different immune cells (Figure 10D) and six different and related immune cells (Figure 10E,F). The immune cells associated with high CBX7 expression include resting mast cells, naïve B cells, and regulatory T cells (Tregs), while the immune cells associated with low CBX7 expression are M0 macrophages, resting NK cells, and activating mast cells. These findings provided substantial confirmation that CBX6 and CBX7 were involved in the control of TME immunological reactivity.

### 3.9. CBX Family Protein Expression in Normal Urothelial Cells (SV-HUC-1) and BLCA Cell Lines (5637 and T24) Verified Using Western Blotting

To validate the cellular expression of CBX family proteins, we detected the protein expression in normal urethral cells (SV-HUC-1) and BLCA cell lines (5637 and T24) (Figure 11A). The expression of CBX1, CBX3, CBX4, CBX5, and CBX8 in the BLCA cell lines 5637 and T24 was higher than that in the normal bladder epithelial cell line SV-HUC-1 (Figure 11B,D–F,I). Compared with normal cell lines, the expression of CBX2 was highly elevated in the BLCA cell line 5637, while no significant change was in the T24 cell line (Figure 11C). The expression of CBX6 in the bladder cancer cell lines 5637 and T24 was lower than that in the normal bladder epithelial cell line SV-HUC-1 (Figure 11G), which was consistent with our previous results, shown in Figure 11F. However, CBX7 expression was higher in the bladder cancer cell line 5637 and lower in T24 than in SV-HUC-1 (Figure 11H).

### 3.10. mIHC Results Confirm the Relationship between CBX6 and CBX7 Expression and Immune Cells

Given that the hypothesized relationship between CBX6, CBX7, and immune cells in the TME may be essential to its function, mIHC was performed on BLCA pathology specimens to further confirm their potential importance. When tumor tissue exhibited low CBX6 expression, the infiltration of regulatory T cells (Tregs) (labelled by CD25) was markedly inhibited in the tumor center. In contrast, T cell regulatory (Treg) recruitment was particularly active in regions with high CBX6 levels (Figure 11D). Moreover, consistent with the preceding findings, the numbers of M1 macrophages (marked by CD86) and CBX6 had strikingly contrasting trends. Therefore, the polarization process of M1 TAMs may be profoundly reduced by active CBX6 expression (Figure 11E). These preliminary findings confirmed our bioinformatics study conclusions that CBX6 was associated with M1 TAM levels and T cell regulatory (Treg) recruitment. When tumor tissue exhibited low CBX7 expression, the infiltration of quiescent mast cells (marked by KIT) in the tumor center was significantly inhibited. In contrast, quiescent mast cell recruitment was particularly active in areas with high CBX6 levels (Figure 11G). Furthermore, consistent with previous findings, the numbers of M0 macrophages (marked by CD14) and CBX7 had clearly opposite trends. Therefore, active CBX7 expression may greatly reduce the recruitment process of M0 cells (Figure 11F). These early results confirm our bioinformatics findings that CBX7 is related to M0 TAM levels and quiescent mast cell recruitment.

## 4. Discussion

The tumor microenvironment (TME) is widely acknowledged as a crucial initial step in cancer therapy [26]. Future research into the mechanism of the TME in specific malignancies, as well as the modification of the TME in tumors, will need new therapeutic targets. The transcriptome data provide credence to the idea that the TME’s immune component helps improve patients’ clinical results [27].

Several forms of cancer have been studied for their potential connection to CBX family protein dysregulation [28]. CBXs have been demonstrated to regulate tumor cell proliferation, survival, invasion, and metastasis, as well as their own creation [29]. Researchers have found a link between CBX proteins and the microenvironment of tumors [30]. However, there has not been a thorough examination of the tumorigenic role of the CBX family in BLCA, particularly the intercellular connection with immune cell infiltration. The current research aimed to determine the prognostic usefulness of CBXs in BLCA by examining their expression profile, protein expression, ROC curve levels, and performing overall survival analysis. The association between CBX6 and CBX7 and immunological checkpoints, m6A, biological functions, patient mutations, immune cell infiltration, and cellular expression were also studied.

The method of post-translational modification controlling CBX proteins, such as phosphorylation and methylation, might account for the discrepancy between the levels of CBX5 mRNA and protein expression [31]. In addition to post-translational regulation of CBX proteins, CBX4 can be used as a SUMO E3 ligase to mediate the recruitment of PRC to methylated histone 3 at K27 (H3K27m3) by promoting sumo protein ligation-related SUMO modification, resulting in transcriptional inhibition of the Gata4/6 gene [32].

The evolution of tumor stage and grade was not simply influenced by the degree of protein expression but was also affected by genetic mutations, the tumor microenvironment, etc. Multivariate Cox analysis showed that CBX5, CBX6, and CBX7 are potential prognostic factors. We observed that CBX6/7 mRNA expression was significantly correlated with the prognosis of BLCA patients. The area under the ROC curve of CBX5 was only 0.5299, and CBX5 mRNA expression was not significantly correlated with the prognosis of BLCA patients, which indicated that CBX5 was not suitable as a molecular marker of BLCA. Therefore, we identified CBX6 and CBX7 as potential prognostic factors for BLCA. The link between high CBX7 mRNA expression and prognosis in bladder cancer was consistent with earlier research [33]. Moreover, increasing evidence suggests that the interaction of immune cells and tumor cells could regulate tumor progression and recurrence, therefore affecting the effect of immunotherapy and clinical outcome [34].

In the results of our analysis, the gene with the highest frequency of somatic mutation in bladder cancer patients was TP53. TP53 is a tumor suppressor gene. Mutation of TP53 can cause cell growth to lose its regulation and then progress towards a tumor. Compared with patients with low CBX6 expression, the greater prevalence of TP53 mutations in individuals with high CBX6 expression may be connected to the worse prognosis of patients with high CBX6 expression. Compared with patients with low CBX7 expression, the frequency of TP53 mutation in patients with high CBX7 expression was lower, which may explain the relatively better prognosis of patients with high CBX7 expression.

M1 macrophages, which originate from monocytic progenitors derived from circulating bone marrow, are one of the most abundant immune cells in cancers [35]. M1 TAMs that have been activated by IFN-, LPS, IL-1, TNF, and GM-CSF are capable of identifying and eliminating malignant cells by phagocytosis and cytotoxicity, as well as producing proinflammatory cytokines that stimulate antitumor immunity [36,37,38]. Tregs are essential for maintaining immunological homeostasis. According to several studies, the infiltration of Tregs into diverse tumor tissues promotes tumor development by reducing antitumor immunity and facilitating tumor immune evasion. In addition, in the tumor microenvironment (TME), Tregs are a diverse subpopulation of cells that express unique immunosuppressive chemicals that promote tumor growth [39,40,41]. Our experimental findings are remarkably compatible with those of the preceding investigations. We think that CBX6 reduces the tumor immune response of patients with bladder cancer by suppressing the polarization of M1 macrophages and by attracting a significant number of Tregs, resulting in a poor prognosis for these patients.

Resting mast cells not only control immune responses but also play a role in the development of a number of inflammatory disorders, including infections, autoimmune illnesses, and cancer [42]. It has been shown in vitro and in vivo that mast cell recruitment increases RCC angiogenesis. Mechanistically, RCC attracts mast cells by modulating PI3K AKT GSK AM signalling, resulting in a poor prognosis [43]. Reduction in ccRCC resting mast cells hampers anticancer immune responses [44]. Tumors having a greater infiltration of CD8+ T cells, M1 macrophages, and CD20+ cells had a better prognosis than those infiltrated with M0/M2 macrophages and high immunological checkpoint protein expression [45]. Studies have indicated that low levels of macrophage M0 are strongly related to a better prognosis in bladder cancer patients [46,47,48]. Our experimental findings are remarkably congruent with those of the preceding investigations. We hypothesize that CBX7 exerts a tumor immune response by suppressing the immunological infiltration of M0 macrophages and attracting a significant number of resting mast cells, resulting in a better prognosis for patients with bladder cancer.

These results showed that CBXs may be crucial regulators in the development of BLCA. The impact of CBX proteins on the immune cell infiltration that regulates the growth of tumors will be explored further.

## 5. Conclusions

This study examined in depth the possible influence of CBXs on BLCA. CBX6 and CBX7 were concurrently associated with overall survival and might have crucial prognostic value in BLCA. CBX6 and CBX7 mRNA expression is correlated with the immunological microenvironment of BLCA. Experiments and cohort studies using clinical samples of CBXs may corroborate the findings.

## Figures and Tables

**Figure 1 diagnostics-13-01393-f001:**
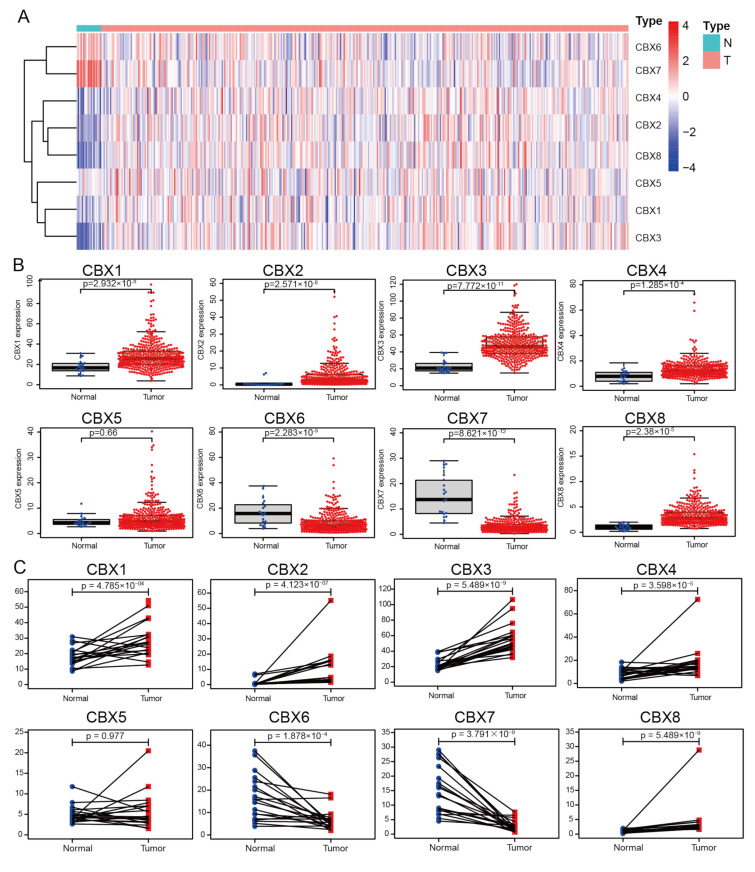
Expression levels of CBX mRNA in BLCA and normal tissues. (**A**) Heatmap depicting CBX mRNA expression in BLCA vs. normal tissues. (**B**) Unpaired samples with 19 normal tissues and 411 BLCA tissues. (**C**) Paired samples comprised 19 normal tissues and their matched BLCA tissues. For statistical analysis, the Wilcoxon rank-sum test was utilized.

**Figure 2 diagnostics-13-01393-f002:**
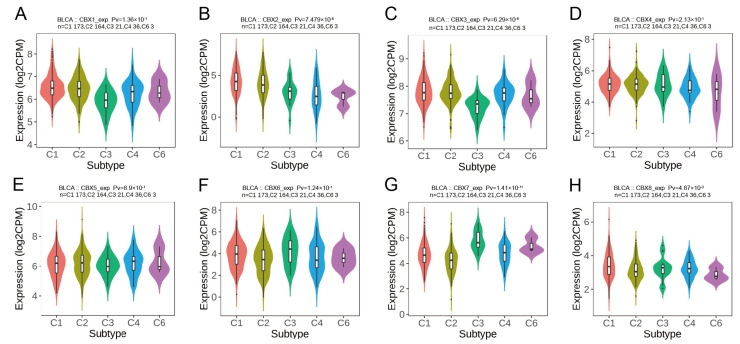
Associations between the expression of CBX mRNA and immunological subtypes throughout BLCA in TISIDB (**A**) CBX1; (**B**) CBX2; (**C**) CBX3; (**D**) CBX4; (**E**) CBX5; (**F**) CBX6; (**G**) CBX7; (**H**) CBX8. C1: wound healing (n = 173); C2: IFN-gamma dominant (n = 164); C3: inflammation (n = 21); C4: lymphocyte depletion (n = 36); C6: TGF-β dominant (n = 3).

**Figure 3 diagnostics-13-01393-f003:**
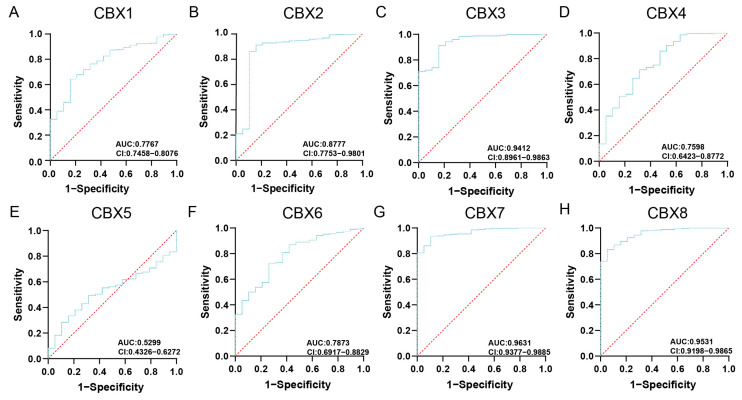
Receiver operating characteristic (ROC) curves for each CBX gene in BLCA. (**A**) CBX1; (**B**) CBX2; (**C**) CBX3; (**D**) CBX4; (**E**) CBX5; (**F**) CBX6; (**G**) CBX7; (**H**) CBX8. CI, confidence interval; AUC, area under the curve.

**Figure 4 diagnostics-13-01393-f004:**
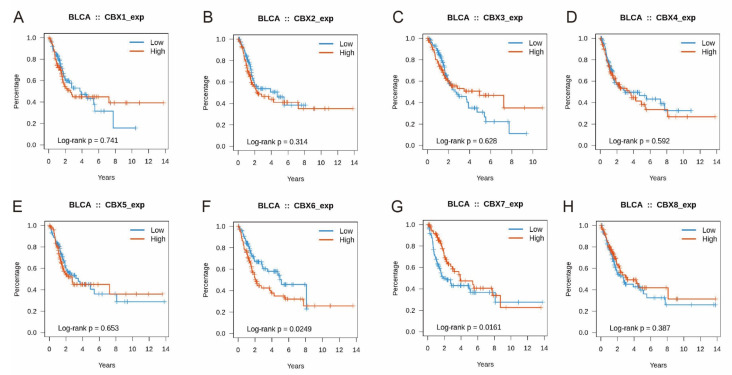
The overall survival rates of CBXs in BLCA were analyzed in TISIDB (**A**) CBX1; (**B**) CBX2; (**C**) CBX3; (**D**) CBX4; (**E**) CBX5; (**F**) CBX6; (**G**) CBX7; (**H**) CBX8. The *p*-values were calculated utilizing the log-rank test.

**Figure 5 diagnostics-13-01393-f005:**
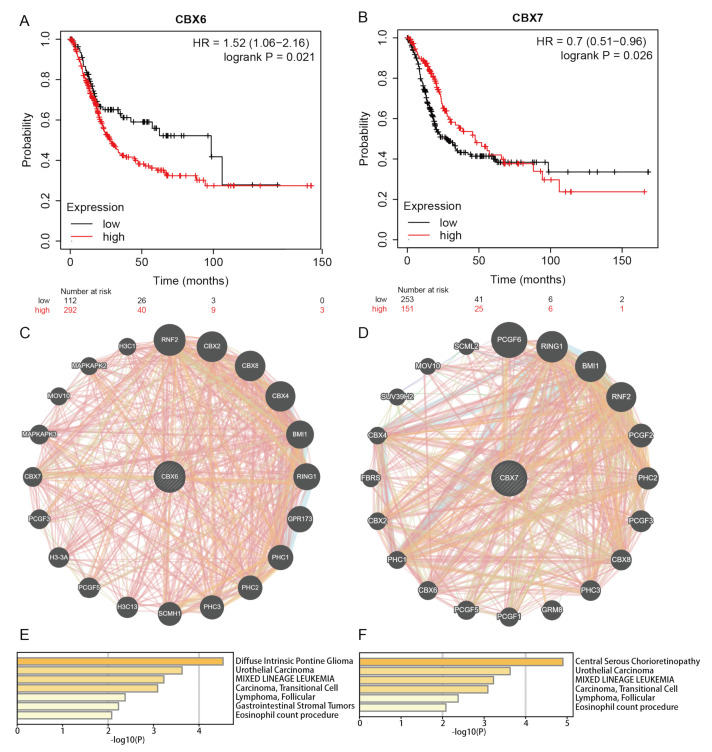
Analysis of the potential roles and pathways of CBX6, CBX7, and their coexpressed neighbors in BLCA. (**A**,**B**) Validation of the predictive value of CBX6 and CBX7 using the Kaplan–Meier plotter database. (**C**,**D**) CBX6, CBX7, and their coexpressed neighbors from GeneMANIA. (**E**,**F**) Analysis of the potential roles and pathways of CBX6 and CBX7 in Metascape.

**Figure 6 diagnostics-13-01393-f006:**
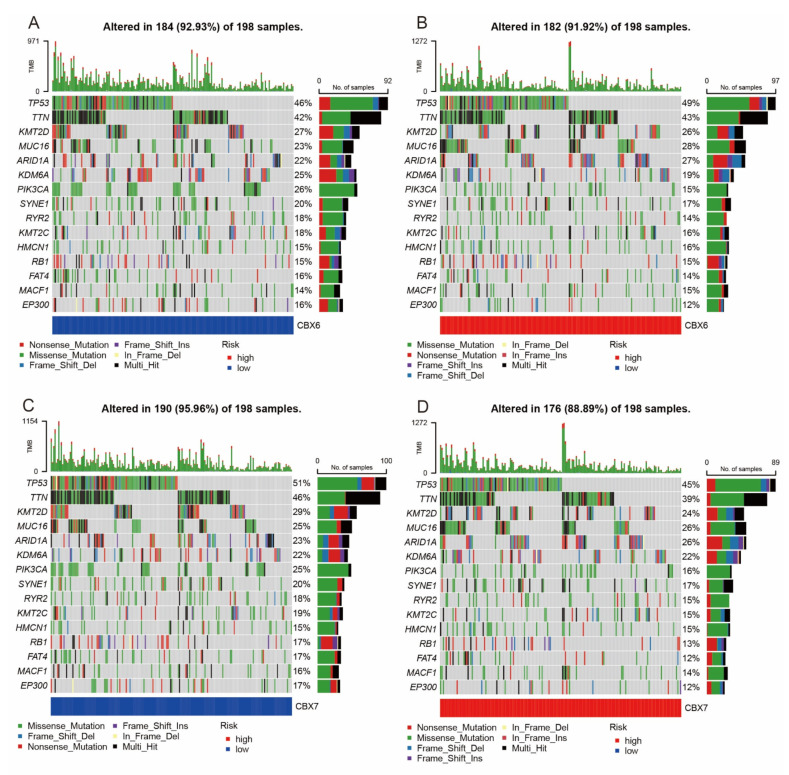
Waterfall diagrams illustrating somatic mutation properties of CBX6 and CBX7. (**A**) CBX6 low-expression group. (**B**) CBX6 high-expression group. (**C**) CBX7 low-expression group. (**D**) CBX7 high-expression group.

**Figure 7 diagnostics-13-01393-f007:**
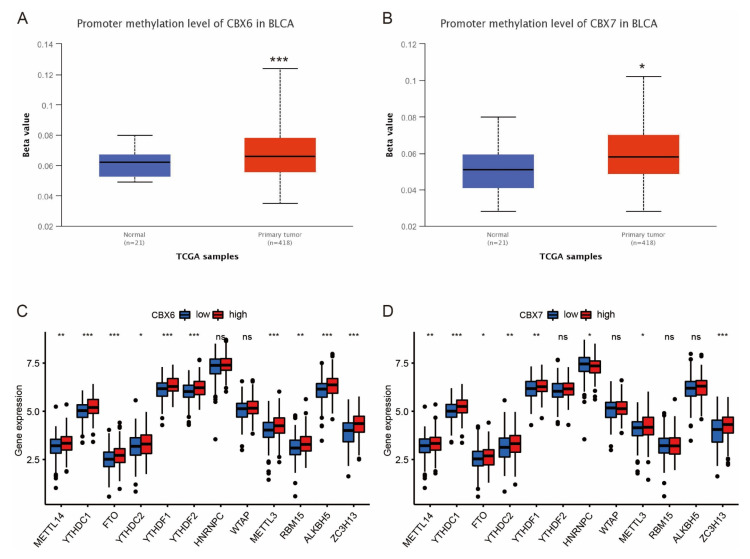
Association of CBX6 and CBX7 with m6A-related gene. (**A**) Promoter methylation levels of CBX6 in normal and bladder cancer tissues. (**B**) Promoter methylation levels of CBX7 in normal and bladder cancer tissues. (**C**) Variations in m6A-related gene expression between groups with high and low CBX6 expression. (**D**) Differences in the expression of m6A-related genes between groups with high and low CBX7 expression. * *p* < 0.05; ** *p* < 0.01; *** *p* < 0.001; ns, not significant.

**Figure 8 diagnostics-13-01393-f008:**
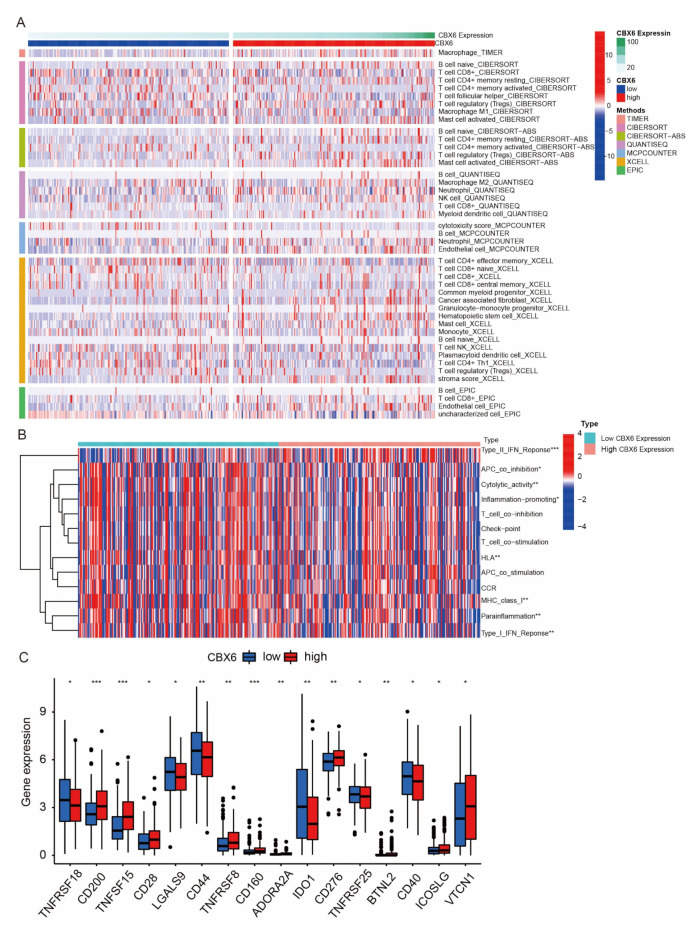
Immune cell infiltration of CBX6 in BLCA patients. (**A**) TIMER, CIBERSORT, CIBERSORTAB, QUANTISEQ, MCPCOUNTER, XCELL, and EPIC algorithms for the immune cell infiltration landscape in the high- and low-risk groups (blue: low CBX6 expression; red: high CBX6 expression). *p* < 0.05 was enforced to ensure that only items with statistically significant differences were reported. (**B**) ssGSEA scores for immune cells and immunological function for groups with high vs. low CBX6 expression. (**C**) Variations in the expression of common immunological checkpoints between CBX6 high- and low-expression groups. * *p* < 0.05; ** *p* < 0.01; *** *p* < 0.001.

**Figure 9 diagnostics-13-01393-f009:**
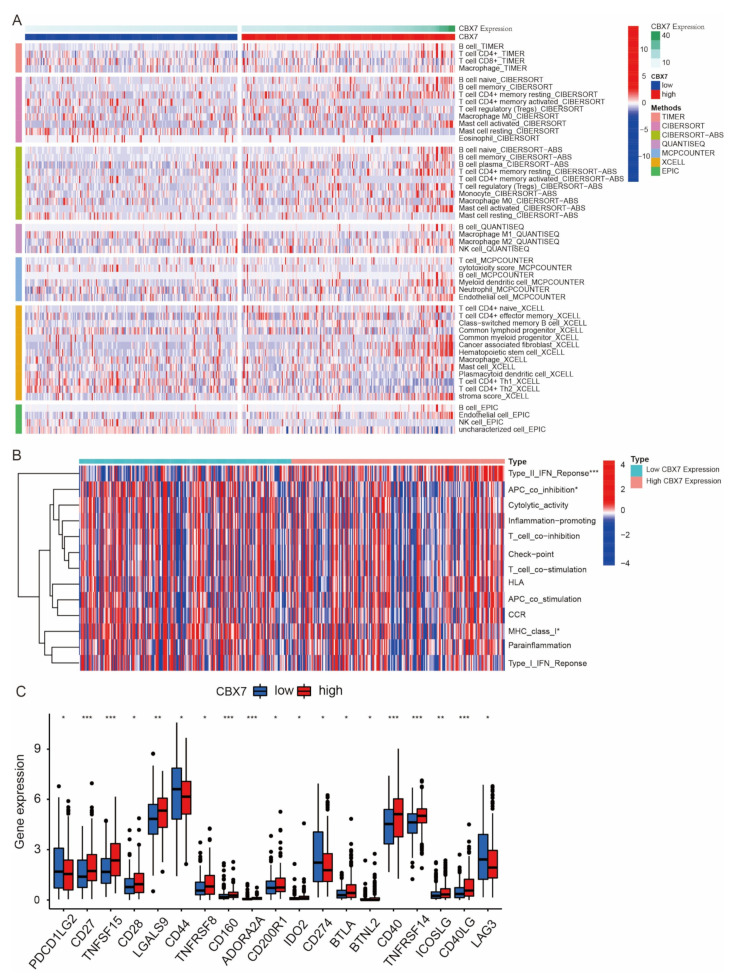
Immune cell infiltration of CBX7 in BLCA patients. (**A**) TIMER, CIBERSORT, CIBERSORTAB, QUANTISEQ, MCPCOUNTER, XCELL, and EPIC algorithms for the immune cell infiltration landscape in the high- and low-risk groups (blue: low CBX7 expression; red: high CBX7 expression). *p* < 0.05 was enforced to ensure that only items with statistically significant differences were reported. (**B**) ss GSEA scores for immune cells and immunological function for groups with high vs. low CBX7 expression vs. low CBX7 expression. (**C**) Variations in the expression of common immune checkpoints between CBX7 high- and low-expression groups. * *p* < 0.05; ** *p* < 0.01; *** *p* < 0.001.

**Figure 10 diagnostics-13-01393-f010:**
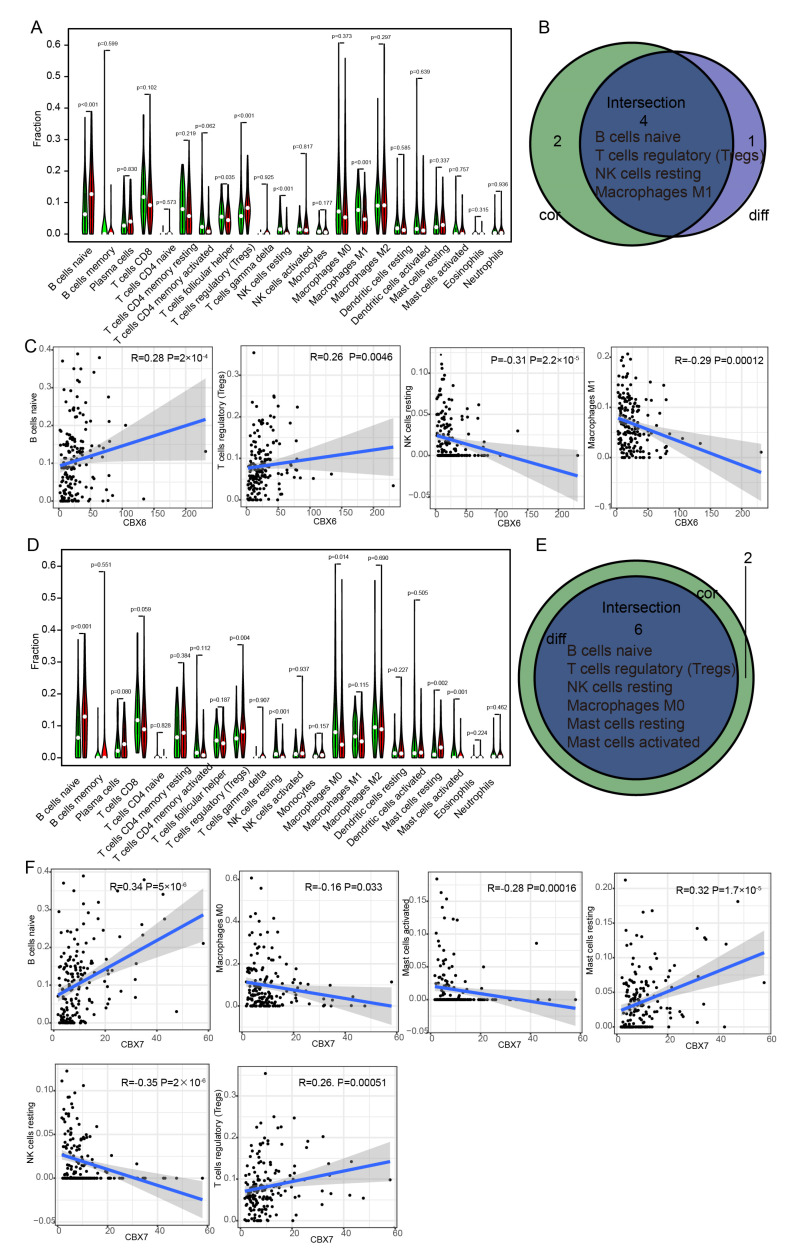
The correlation between CBX6/CBX7 and immune cells. (**A**,**D**) The differential expression of immune cell types in the CBX6 (**A**) and CBX7 (**D**) high- and low-expression groups. (**B**,**E**) Overlap between the correlation and differential analyses of CBX6 and CBX7. (**C**,**F**) Correlation analysis of the expression of CBX6 and CBX7 and the levels of different immune cells.

**Figure 11 diagnostics-13-01393-f011:**
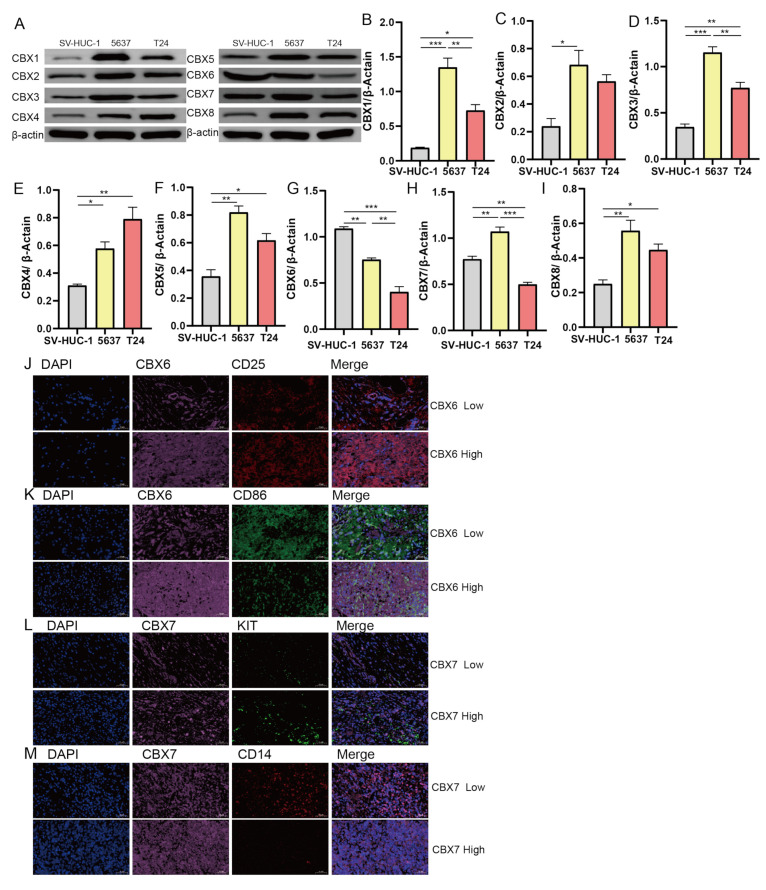
Validation of CBX family protein expression levels in BLCA cell and tissue samples. (**A**) The expression levels of CBX family proteins were detected by Western blotting in normal urothelial cell lines (SV-HUC-1) and BLCA cell lines (5637 and T24) (n = 3). (**B**–**I**) Quantitative results of (**A**). (**J**,**K**) Expression levels of CBX6, CD25, and CD86 in BLCA tumor tissue samples analyzed by immunohistochemistry (IHC). (**L**,**M**) Expression level of CBX7, KIT, and CD14 analyzed by IHC in BLCA tumor tissue samples. * *p* < 0.05; ** *p* < 0.01; *** *p* < 0.001.

**Table 1 diagnostics-13-01393-t001:** The multivariable Cox regression model of CBXs and clinical factors analyzed by TIMER.

	Coef	HR	95% CI_1	95% CI_u	*p*-Value	Sig
Age	0.035	1.035	1.018	1.053	0.000	***
gendermale	−0.201	0.818	0.574	1.165	0.266	ns
raceBlack	0.57	1.769	0.712	4.394	0.219	ns
raceWhite	0.014	1.014	0.495	2.079	0.969	ns
stage2	14.538	2,060,365	0	Inf	0.994	ns
stage3	15.118	3,678,483	0	Inf	0.993	ns
stage4	15.656	6,299,742	0	Inf	0.993	ns
Purity	0.21	1.234	0.598	2.545	0.569	ns
CBX1	−0.156	0.855	0.621	1.178	0.339	ns
CBX2	−0.081	0.922	0.78	1.089	0.340	ns
CBX3	−0.235	0.79	0.546	1.144	0.212	ns
CBX4	0.051	1.052	0.758	1.461	0.762	ns
CBX5	0.284	1.328	1.059	1.666	0.014	*
CBX6	0.3	1.35	1.125	1.619	0.001	**
CBX7	−0.409	0.664	0.536	0.824	0.000	***
CBX8	0.019	1.02	0.729	1.426	0.910	ns

Coef: a regression coefficient; HR: hazard ratio; 95% CI: 95% confidential interval. * *p* < 0.05; ** *p* < 0.01; *** *p* < 0.001; ns, not significant.

## Data Availability

Any interested party is welcome to seek access to the datasets used in this work by contacting the corresponding authors.

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
