# Peer review of "Identification of the Prognostic Biomarkers CBX6 and CBX7 in Bladder Cancer"

_diagnostics, 2023, doi:10.3390/diagnostics13081393_

Round 1

Reviewer 1 Report

In this study, the correlation between the CBX family and BLCA patient prognosis was investigated to analyze CBX6 and CBX7 as putative prognostic markers. Data were taken mostly by platforms online available such as  ONCOMINE, The Cancer Genome Atlas database, and Proteinatlas. This means that the initial statements made are based on data sets from others. However, finally, the main results were partly confirmed by expression analysis of one normal urethral cells (SV-HUC-1) in contrast to two cancer cell lines (5637 and T24) by western blotting and immunohistological analysis.

Major issues

-          2.11. Cell lines: Have the cell lines been authenticated and tested for mycoplasma?

-          2.12. Western blotting: How much soluble protein (µg) was loaded on the gels? Which kind of gels were used? How the loading control and the internal standards were performed? Which secondary antibody was used?

-          2.13. Multiple immunohistochemistry: From where the tissue sections were received? Which antibodies were used and from where they were purchased? Ethics vote?

-          3.4. Prognostic value of CBXs in BLCA patients: CBX1 and CBX3 seem to have a greater influence on the overall survival.

-          Figure 11. How many technical and biological replicates were analyzed by western blotting? Please add to the figure legend.

-          The discussion should not be a repetition of the results section. A critical discussion with the current literature and data is desirable.

Minor issues:

-          British or American English: make it consistent.

-          Line 41: 50 - 70 years old (Spaces before and after the hyphen)

-          Line 84: f BLCA. and provide (point or comma?)

Reviewer 2 Report

Major comments:

1. how do the authors explain the difference in CBX expression between deiffrent UC cell lines? 

2. The authors need to present the expression of all CBX family, not just CBX6 and CBX 7

3. the survival data is only deflected in a univariate kaplan meier analysis - the authors need to perform a multivariate cox analysis including all known clinical prognostic and histopathological factors for UC patients and assess whether q-values (not just p-values) from TCGA data are actually significant. 

Round 2

Reviewer 1 Report

All issues have been addresed.

Reviewer 2 Report

The authors have successfully addressed most of the Reviewers' comments. No additional concerns at this point.